# Predictors of Response to a Medial Branch Block: MRI Analysis of the Lumbar Spine

**DOI:** 10.3390/jcm8040538

**Published:** 2019-04-19

**Authors:** Jun-Young Park, Doo-Hwan Kim, Dong-Kyun Seo, Syn-Hae Yoon, Gunn Lee, Sukyung Lee, Chan-Hye Park, Sung Eun Sim, Jeong-Hun Suh

**Affiliations:** 1Department of Anesthesiology and Pain Medicine, Asan Medical Center, University of Ulsan College of Medicine, Seoul 05505, Korea; anesthesia.pains@gmail.com (J.-Y.P.); knaaddict@gmail.com (D.-H.K.); shotguny@hanmail.net (G.L.); may1286@hanmail.net (S.L.); chanhye718@gmail.com (C.-H.P.); 2Department of Anesthesiology and Pain Medicine, Jun Orthopedic Hospital, Seoul 04790, Korea; dongkooseo@gmail.com; 3Department of Anesthesiology and Pain Medicine, National Police Hospital, Seoul 05715, Korea; synhae@gmail.com; 4Department of Anesthesiology and Pain Medicine, Seoul ST. Mary’s Hospital, The Catholic University of Korea, Seoul 06591, Korea

**Keywords:** medial branch block, diagnostic medial branch block, facet joint pain, facet joint syndrome, facet joint tropism, facet angle, facet angle difference, intervertebral disc degeneration, low back pain

## Abstract

The aim of this study was to determine the association between radiologic spinal pathology and the response to medial branches block (MBB). This retrospective observational study compared 165 patients. A successful response was defined as ≥30% or a 2-point reduction in the numeric rating scale (NRS) compared with the baseline at the 1-month follow-up. The facet angle, facet angle difference, facet joint degeneration, disc height and spondylolisthesis grade were analyzed from an MRI at the L3 to S1 levels. Univariate and multivariate logistic regression analyses were used to evaluate independent factors associated with a successful response of MBB. In the univariate analysis, the disc height at L5–S1 and facet angle difference at L3–4 were lower in the positive responders (*p* = 0.022 and *p* = 0.087, respectively). In the multivariate analysis, the facet angle difference at L3–4 and disc height at L5–S1 were independent factors associated with a successful response (odds ratio = 0.948; *p* = 0.038 and odds ratio = 0.864; *p* = 0.038, respectively). In patients with a degenerative disc at L5–S1, MBB can lead to a good response for at least one month. In patients with facet tropism at L3–4 level, the response to MBB after one month is likely to be poor.

## 1. Introduction

Low back pain is usually a self-limiting symptom, but is a very common problem [1,2,3]. The lifetime prevalence of low back pain is known to be as high as 84%, and the prevalence of chronic low back and leg pain may be 23%, related to 11–12% of the patients being disabled [2,4]. The cause of lower back pain is often multifactorial, with there being various causes of pain in the lumbar spine [5]. These include problems with intervertebral discs, lumbar facet joints, sacroiliac joints, ligaments, fascia, muscles and spinal nerves [6,7]. The facet joint has been increasingly implicated as an important source of pain [6]. Pain can arise from any structure within the facet joint complex, including the fibrous capsule, synovial membrane, hyaline cartilage and bone [6]. Lumbar facet arthropathy is a common radiographic finding and is suggested to account for between 15% and 40% of low back pain cases [1,8,9].

The relationship between the paired facet joints and intervertebral discs, which form the basic anatomical unit referred to as the three-joint complex of the spine, is one of considerable complexity [8]. The facet joints typically bear between 3% and 25% of the relevant axial load under normal conditions [8]. Previous studies show that pathologic conditions, such as facet degeneration, degenerative disc disease and spondylolisthesis, are associated with a loading imbalance in the three-joint complex. This imbalance can aggravate degenerative change to the facet joint and intervertebral disc. Several studies attempted to identify associations between the facet angle and degenerative change to the facet joint or intervertebral disc [10,11]. Moreover, facet tropism, where the facet angle shows a difference between the right and left sides, has also been studied to determine the cause of facet joint-induced low back pain [10,11,12,13].

Various studies show that neither historical nor physical examination findings, nor radiologic facet joint pathology, can reliably be used to diagnose a painful facet joint [14,15,16,17,18]. Although there is still some controversy surrounding the appropriate management of lumbar facet joint pain, the diagnostic facet joint nerve block is known to have level I or II-1 evidence [19]. In clinical practice, it is accepted that a response to the blockade of either the facet joints themselves, or, more frequently, the medial branches that innervate them, is the most reliable means for diagnosing facet joints as the cause of low back pain [20]. However, there are only a few studies that factored in the response to a facet joint block or medial branch block (MBB) [21,22,23]. Moreover, no study to date has examined the relationships between degenerative spinal pathology such as facet tropism, facet angle, facet degeneration and degenerative intervertebral disc, and the response to a diagnostic MBB. Therefore, the purpose of this study was to analyse the associations between radiologic spinal pathology and the response to MBB.

## 2. Materials and Methods

This single-centre retrospective observational study used the institutional registry records of 252 patients who experienced MBB between January 2010 and October 2015. The ethics board of our institution approved this study (approval number, S2016-1537-0001). Under the approval of the ethics board, we collected the data from the electronic medical record.

### 2.1. Patients

Adult patients with back pain who underwent magnetic resonance imaging (MRI) prior to MBB were included in the study. The enrolled patients were suspected of facet joint syndrome because of axial pain, absence of radiculopathy and paravertebral tenderness.

Patients were excluded from the study if there was a history of recent trauma affecting the lower back pain, and prior spine surgery including discectomy or fusion. Other anatomical abnormalities associated with the spine, such as a compression fracture, facet joint cyst or metastatic cancer, were also excluded. Patients lost to follow-up were also excluded from the study.

### 2.2. Interventions

All procedures were performed by an expert, who was board certified in anaesthesiology and pain medicine. All treatments were performed as an outpatient setting. Patients received bilateral medial branch nerve blocks of the L2, L3, L4 and L5 dorsal rami, according to the standard routine performed in our centre. Every patient underwent MBB in a sterile operating room. The procedure was performed on the prone-position using a posterior approach. Following a sterile skin preparation and placement of a fenestrated sterile drape, a 25-gauge 3.5-inch spinal needle was used for each injection. At the level L2 to L4, MBBs are performed by targeting the junction of the upper border of the transverse process and superior articular process under fluoroscopy guidance. The L5 dorsal ramus is also blocked in the groove between the ala of the sacrum and the superior articular process of S1 under fluoroscopy guidance [24]. The injectate consisted of 1 mL of 1% lidocaine with 1 mg of triamcinolone and was injected at each level. Because bilateral MBB was performed at the L2, L3, L4 and L5 dorsal rami, the total volume of the injectate was 8 mL in each patient.

### 2.3. Outcome Measures and Follow-Up

The baseline characteristics, such as age, gender, body weight, height and baseline pain intensity, were obtained. The clinical data, consisting of a medical interview and pain assessment, were collected at the baseline and the one-month follow-up. An 11-point numeric rating scale (NRS) was used to assess the pain intensity [25]. All patients were asked to consider and rate the average severity of their symptoms over the previous week. A successful response of MBB was defined as a reduction of ≥ 30% or 2 points in the NRS pain intensity at the one-month follow-up examination, in comparison with the baseline score [26]. If present, complications during the procedure were reported, and adverse events were further evaluated at the monthly follow-up visits.

### 2.4. Measurements

The angles and lengths were measured on axial T2-weighted spin-echo imaging from L3–4 through to L5–S1 levels. The facet angle, facet angle difference and facet joint degeneration grade were measured on axial images. The axial sections were selected according to the traversal of the intervertebral discs or the facet joint at each level. The disc height and spondylolisthesis were measured on sagittal images. The sagittal sections were selected according to the midline of the spine. All data were digitally measured using the Asan Medical Center PACS system (PetaVision, version 2.1, Seoul, Korea).

On an axial scan, one line was drawn along the midline of spinous processes, and then further lines were drawn through each facet joint, tangential to the superior articular process. Software on the PACS system was then used to calculate the left and right facet joint angles subtended by each of the oblique lines and the sagittal plane (Figure 1). The mean angle was calculated from the right and left facet angles at each level. The facet angle differences were also calculated as the difference between the right and left facet angles at each level [27,28]. Facet joint degeneration was classified according to validated grading systems for facet joint osteoarthritis [10,29]. Grade 0 (normal) indicates a normal facet joint, whereas grades 1–3 display increasing signs of facet joint degeneration. Grade 1 (mild) shows a narrowing of the joint space, small osteophytes or mild hypertrophy of the articular process. Grade 2 (moderate) demonstrates moderate osteophytes, moderate hypertrophy of the articular process or a mild subarticular bone erosion. Grade 3 (severe) include large osteophytes, severe hypertrophy of the articular process, subarticular bone erosion or subchondral cysts [10,29].

The anterior and posterior disc heights were obtained from the midsagittal scan, with the disc height being calculated as the average of the anterior and the posterior disc heights. Spondylolisthesis was categorized on the basis of the ratio of the overhanging part of the superior vertebral body to the anteroposterior length of the adjacent inferior vertebral body, as follows: 1 = 0% to 25%; 2 = 25% to 50%; 3 = 50% to 75%; 4 = 76% to 100%; and 5 = above 100% [30]. All imaging parameters, such as the facet angle, facet angle difference, facet degeneration grade, facet fluid, spondylolisthesis, and disc height of each patient, were measured three times, and the means of the results was calculated by two experienced pain physicians.

### 2.5. Statistical Analysis

The continuous variables are presented as means with standard deviation (SD) or medians with the interquartile range (IQR), if skewed. The categorical variables are presented as absolute numbers and percentages.

To analyse differences between positive and negative responders at one month post-MBB, continuous variables were compared using the Student’s *t*-test or the Mann–Whitney *U*-test, as appropriate. The categorical data were compared using the chi-square test to assess differences between the two groups. The most relevant factors associated with a successful response at one month post-MBB based on biological plausibility, clinical importance and statistical considerations were included in a univariate logistic regression analysis. The inclusion of variables into the final multivariate logistic regression analysis to evaluate independent factors associated with a successful response at one-month post-procedure was based on the biological plausibility, clinical importance, multicollinearity, and statistical considerations (*p* < 0.10). To analyse the correlation between the percentage of improvement on the NRS, and the facet angle difference at L3–4 and disc height at L5–S1, a Spearman’s rank correlation analysis was performed. Statistical analyses were performed using SPSS 21.0 for Windows (SPSS Inc., Chicago, IL, USA). A two-tailed *p*-value of < 0.05 was considered to indicate a statistically significant difference.

## 3. Results

A series of 252 patients who had been subjected to an MBB and who underwent MRI prior to MBB between January 2010 and October 2015 were screened for their eligibility to participate. Fifty-two patients were excluded because of previous spine surgery, and 35 patients were excluded due to being lost to follow-up. In total, 165 patients fulfilled both the inclusion and exclusion criteria (Figure 2).

The baseline patient demographic characteristics are shown in Table 1, categorized into negative and positive responders. There were 66 (40%) negative responders and 99 (60%) positive responders. None of the baseline characteristics differed between the positive and negative responder groups.

Table 2 compares the MRI characteristics of the negative and positive responders at one month post-MBB. The disc height at L5–S1 was significantly lower in the positive responders than in the negative responders (*p* = 0.022). The facet angle difference at L3–4 was smaller in the positive responders than in the negative responders, although this difference failed to reach a statistical significance (*p* = 0.087). No other MRI-measured characteristics differed significantly between the two groups. In the univariate logistic regression analysis, the facet angle difference at L3–4 and L5–S1, disc height at L3–4 and L5–S1, and facet fluid at L5–S1 were analyzed. The univariate logistic regression analysis showed that the facet angle difference at L3–4 and disc height at L5–S1 were significantly associated with a positive response at one month. In view of the biological plausibility, clinical importance, multicollinearity, and statistical considerations, the facet angle difference at L3–4, disc height at L5–S1, and facet fluid at L5–S1 were analysed in a multivariate logistic regression analysis. We confirmed that there was no multicollinearity between the variables. The multivariate logistic regression analysis showed that the facet angle difference at L3–4 (odds ratio = 0.948; *p* = 0.038) and disc height at L5–S1 (odds ratio = 0.864; *p* = 0.038) were independent factors significantly associated with a successful response at one-month post-procedure (Table 3). The facet fluid at L5–S1 was also independently associated with a successful response to MBB, although it showed only a marginal significance (odds ratio = 0.785; *p* = 0.056; Table 3). The Spearman’s rank correlation analysis reveal a weak but statistically significant correlation between the NRS difference, NRS change and disc height at L5–S1 (r = −0.209; *p* = 0.017 and r = −0.206; *p* = 0.019, respectively; Appendix A). Also, there was a marginal statistically significant correlation between the NRS change and facet angle difference at L3–4 (r = 0.163; *p* = 0.062).

## 4. Discussion

Low back pain is a common global health problem [4]. It frequently limits activity and creates absence from work, thereby resulting in a substantial economic burden [31]. The anatomic cause of low back pain has been known to be induced by various structures, such as the intervertebral discs, lumbar facet joints, sacroiliac joints, ligaments, fascia, muscles, or spinal nerves [6,7]. Facet joint pain is a common cause of chronic low back pain, accounting for 5–90% of cases [8]. The lumbar facet joints and intervertebral disc compose the three-joint complex of the spine [8]. The lumbar facet joints form the posterolateral articulations of the spine, connecting the vertebral arch of one vertebra to the arches of the vertebrae below and above. Like true synovial joints, facet joints have a distinct joint space containing joint fluid (1–1.5 mL), a synovial membrane, hyaline cartilage surfaces and a fibrous capsule [6]. In many cases, facet joint pain is the result of accumulated strain or trauma over the course of a lifetime, which can aggravate facet joint degeneration [7,8,32,33]. With repetitive strain and inflammation, the synovial facet joints can fill with fluid and distend, resulting in pain from the stretching of the joint [7,8,33].

Numerous studies show that degenerative or anatomical changes of the spine, such as degenerative disc disease, spondylolisthesis, facet hypertrophy, facet angle and facet tropism, are associated with facet join pain or facet degeneration [11,18,21,34,35,36,37]. Some, but not all of these studies, found that the facet angle was associated with facet joint disease [13,38,39,40]. Moreover, facet tropisms, defined as asymmetry in both of the facet joint angles, are known to be the potential anatomical predisposing factors for lumbar degenerative changes [10,13]. Although it is a subject of debate [10,34,41], some researchers argue that facet tropism is associated with facet joint disease [34].

In this study, the facet angle difference at L3–4 and disc height at L5–S1 were independent factors significantly associated with a successful response at the one-month follow-up examination. In other words, facet tropism at the L3–4 level is associated with a poor response to MBB. Degenerative change of the disc at L5–S1 is associated with a good response to MBB. There are several possible explanations for the effective pain relief observed after MBB in patients with a low intervertebral disc height at L5–S1, and with those with a reduced facet angle difference at L3–4. In a finite element analysis, Park et al. found that disc degeneration correlated with facet joint force [42]. As the joint and intervertebral disc is a three-joint complex, degenerative change of the disc can affect facet joint loading. Like the effect of another block to a loaded structure, MBB may be effective for increasing facet loading accompanied with disc degeneration. Also, it is possible that injectate from the MBB procedure spread to other structures, such as back muscles or intervertebral discs, which may have been causing axial pain, and this may have affected the response to MBB. However, the volume of injectate used was small, and the MBB procedure was performed under fluoroscopic guidance. We therefore think that it is unlikely that the injectate spread to the intervertebral disc or other structures. Moreover, the MBB response was only significantly different in those patients with an L5–S1 degenerative disc, not in degenerative discs at other levels. It therefore seems unreasonable to consider the effect as being due to the action of the injectate on other sites.

We found that facet tropism at L3–4 was associated with a poorer response to MBB, and that disc degeneration at L5–S1 was associated with a better response to MBB. As this study was of a retrospective nature, MBB was performed according to the standard routine performed in our centre using conventional oblique angle fluoroscopy, without taking facet tropism into consideration. Therefore, it is possible that MBB was not performed on the proper target. The target of MBB is the medial branch of the lumbar dorsal ramus. In cases of facet tropism, the appropriate needle position may be difficult to achieve under the conventional oblique angle of fluoroscopy. Another possibility is that MBB may have been less effective for direct facet loading caused by facet tropism. As mentioned above, the association between facet joint pain and facet tropism is controversial. However, facet tropism appears to make the corresponding segment more vulnerable to external moments or an anterior shear force [13]. Increased facet contacting force resulting from facet tropism may result in poor MBB responsiveness, regardless of facet degeneration [10,34,40]. Moreover, because of the sacrum and the lower lumbar segment being relatively more fixed than the upper lumbar segment, it could be that the facet joint contact force induced by facet tropism was higher in the upper than lower segment. There is little that is known about the differences in MBB responses at different spinal levels and their association with aetiology, such as facet tropism or degenerative disc disease. Further research is required to better understand the different MBB responses associated with L3–4 facet tropism and L5–S1 disc degeneration.

In this study, we found no association between facet degeneration and the response to MBB. It is known that facet joint pain is induced by facet degeneration and that MBB is effective for such pain [5,7,8,20]. Our findings suggest that facet loading induced by tropism or a degenerative disc is a more important factor for attaining a positive MBB response than facet degeneration. Furthermore, we found a weak but statistical correlation between the NRS difference, NRS change and disc height at L5–S1. Aslo, there was a marginal statistically significant correlation between the NRS change and facet angle difference at L3–4. In clinical practice, a response to MBB is known to be a reliable means for diagnosing facet joints. In view of these considerations, a binary outcome such as responder or non-responder is likely to more compatible with a suspicion of facet joint pain than a continuous outcome such as an NRS difference or NRS change.

As far as we are aware, this is the first study to examine facet tropism and degenerative disc disease as factors for attaining an MBB response. Numerous studies outline the limitations of diagnosing facet joint pain through the examination of patient history, physical examinations and radiologic findings, and they conclude that an analgesic response to an image guided intra-articular block or MBB is the only reliable and valid method for identifying facet joint pain [7,8,14,17,20,33,43]. However, there are few studies on the effects of the MBB response [21,22,23]. Moreover, no studies to date have examined the relationship between degenerative spinal pathology such as the facet tropism, facet angle, facet degeneration or degenerative intervertebral disc, and the response to a diagnostic MBB. In this study, we assumed that degenerative or anatomical changes to the spine, such as facet tropism or degenerative disc disease, might increase the strain on the facet joints and cause pain, thereby affecting the response to the MBB. We evaluated this hypothesis by correlating MRI findings such as the facet angle, tropism, degeneration, spondylolisthesis and disc degeneration with the response to MBB.

There are several limitations to this study. This study was a retrospective review and did not employ controls, blinding or randomization. Although we analysed the imaging parameter as the average of the values measured three times, there may be a possibility that there is a problem of inter-rater reliability or intra-rater reliability. We lost 35 patients from follow-up; they were excluded from this study on the basis of a retrospective chart review. We also did not evaluate other clinical outcomes, such as the Oswestry Disability Index (ODI) or patient global impression of change (PGIC). Although other studies also only examined outcomes according to the NRS [21], the use of a variety of outcomes would have been beneficial [21,44]. The definition of a successful response may also be criticized, as different results might have been obtained with a different definition, although our definition of a successful response was established according to previous studies and recommendations [26,45]. Although facet tropism at L3–4 level and disc degeneration at L5–S1 are independent factors associated with the response to MBB, and although this was statistically significant, the association may not be large enough. Despite these limitations, the results are clinically relevant. When performing MBB, we are likely to expect a poorer response in patients with facet tropism at the L3–4 level, and a better response in patients with degenerative disc disease at the L5–S1 level. These findings also give hope for the management of complex degenerative changes affecting the lumbosacral spine, especially in the elderly. Oftentimes, such patients present with axial pain due to abnormal facet loading at the lumbosacral junction, so that facet arthropathies may not be an isolated phenomenon and are commonly found together with evidence of synovial cysts and degenerative scoliosis [46,47,48]. Therefore, the minimal invasiveness of MBB and the positive results obtained, especially in degenerative disc disease at the L5–S1 level, would indicate the rationale for further clinical studies.

## 5. Conclusions

The facet angle difference at L3–4 and disc height at L5–S1 may be independent factors associated with a successful response to MBB in our study. The facet tropism at the L3–4 level may be associated with a poor response of MBB after one month. A degenerative disc disease at the L5–S1 level may be associated with a good response of MBB after one month.

## Figures and Tables

**Figure 1 jcm-08-00538-f001:**
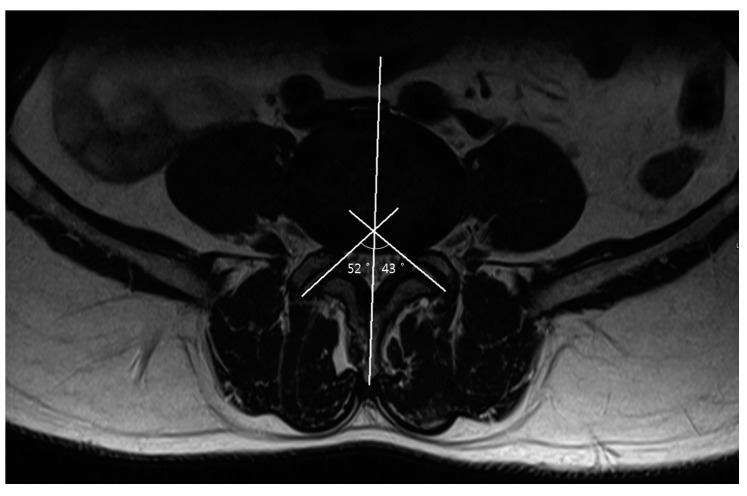
Measurement of the facet angle and facet tropism at L4–5. The facet angles are the angles between the line drawn through the midsagittal plane of the vertebra and the lines through each facet joint tangential to the superior articular process. The facet angle difference was calculated as the difference between the right and left facet angle.

**Figure 2 jcm-08-00538-f002:**
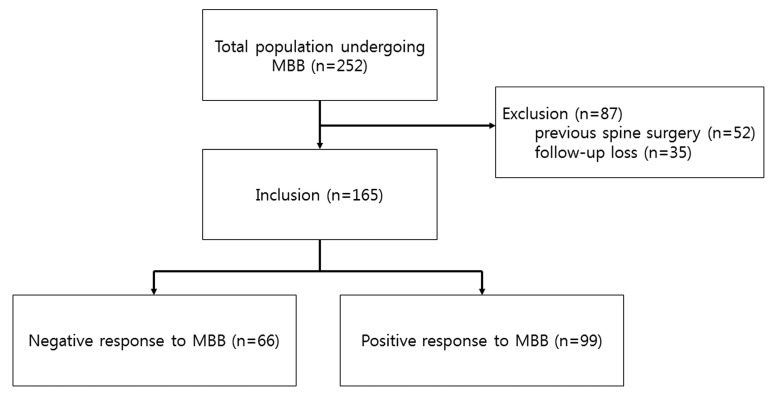
Flow diagram of the study procedure. MBB = Medial branch block.

**Table 1 jcm-08-00538-t001:** Baseline characteristics of the patients.

	Negative Response to MBB (*n* = 66)	Positive Response to MBB (*n* = 99)	*p*-Value
Age (year) (median, IQR)	63.0 (53.0–72.0)	64.0 (57.0–71.0)	0.454
Gender (male/female) (number, %)	17/49 (25.8/74.2%)	34/65 (34.3/65.7%)	0.303
Weight (kg) (median, IQR)	60.7 (53.4–68.8)	61.3 (55.0–68.0)	0.680
Height (cm) (median, IQR)	158.2 (152.4–165.0)	157.0 (152.0–164.4)	0.854
BMI (kg/m^2^) (median, IQR)	24.63 (18.38–30.27)	25.10 (19.51–30.42)	0.927
MQS (points) (median, IQR)	8.00 (4.80–9.60)	8.00 (4.15 –9.00)	0.216
Pain intensity (NRS) (mean ± SD)	6.37 ± 1.85	6.64 ± 1.69	0.366

Data are presented as mean ± SD, median (IQR), or number (%) as appropriate. MBB = Medial branch block, IQR = interquartile range; NRS = numeric rating scale; BMI = body mass index; MQS = Medication quantification scale; NRS = numeric rating scale; SD, standard deviation.

**Table 2 jcm-08-00538-t002:** Characteristics of the negative and positive responders after the medial branch block.

	Negative Response to MBB (*n* = 66)	Positive Response to MBB (*n* = 99)	*p*-Value
Facet angle (°) (mean ± SD)			
L3–4	38.82 ± 10.74	37.07 ± 10.69	0.304
L4–5	45.15 ± 9.81	45.08 ± 9.68	0.661
L5–S1	47.88 ± 9.04	49.44 ± 11.97	0.345
Facet angle difference (°) (mean ± SD)			
L3–4	7.42 (2.09–14.70)	6.02 (2.54–9.26)	0.087
L4–5	7.38 (3.00–12.76)	6.54 (3.27–11.10)	0.313
L5–S1	9.20 (5.00–16.57)	7.78 (4.33–11.18)	0.102
Facet degeneration grading (number, %)			
L3–4 (Gr 0/1/2/3)	15 (22.7%)/31 (47.0%)/12 (18.2%)/8 (12.1%)	21 (21.2%)/51 (51.5%)/21 (21.2%)/6 (6.1%)	0.551
L4–5 (Gr 0/1/2/3)	14 (21.2%)/25 (37.9%)/22 (33.3%)/5 (7.6%)	14 (12.1%)/42 (42.4%)/39 (39.4%)/6 (6.1%)	0.432
L5–S1(Gr 0/1/2/3)	15 (22.7%)/26 (39.4%)/18 (27.3%)/7 (10.6%)	21 (21.4%)/41 (41.8%)/31 (31.6%)/5 (5.1%)	0.578
Facet fluid (mm) (median, IQR)			
L3–4	1.11 (0.77–2.22)	0.98 (0.80–1.44)	0.443
L4–5	1.13 (0.79–2.46)	1.10 (0.65–1.77)	0.353
L5–S1	0.98 (0.65–2.10)	0.98 (0.54–1.29)	0.189
Spondylolisthesis (number, %)			
L3–4	9 (13.6%)	14 (14.1%)	0.927
L4–5	14 (21.2%)	15 (15.2%)	0.405
L5–S1	6 (9.1%)	14 (14.2%)	0.531
Disc height (mm)			
L3–4	7.85 (6.65–9.06)	7.55 (6.09–8.90)	0.198
L4–5	8.21 (6.56–9.72)	8.20 (6.62–9.87)	0.713
L5–S1	9.94 (7.51–11.26)	8.73 (6.70–10.31)	0.022

Data are presented as mean ± SD, median (IQR), or number (%) as appropriate. MBB = Medial branch block, SD, standard deviation; IQR = interquartile range; Gr = Grade.

**Table 3 jcm-08-00538-t003:** Logistic regression analysis of factors associated with a positive response after a medial branch block.

Variables	Univariate Analysis				Multivariate Analysis			
	Coefficient	OR	95% CI	*p*-Value	Coefficient	OR	95% CI	*p*-Value
Facet angle difference (°)								
L3–4	−0.06	0.94	0.90–0.99	0.020	−0.05	0.95	0.90–0.10	0.038
L5–S1	−0.04	0.96	0.93–1.00	0.080				
Disc height (mm)								
L3–4	−0.12	0.89	0.75–1.05	0.164				
L5–S1	−0.15	0.86	0.75–0.98	0.022	−0.15	0.86	0.75–0.99	0.038
Facet fluid (mm)								
L5–S1	−0.22	0.81	0.63–1.02	0.078	−0.24	0.79	0.61–1.01	0.056

OR = odds ratio; CI = confidence interval.

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
