# Peer review of "Predictors of Response to a Medial Branch Block: MRI Analysis of the Lumbar Spine"

_jcm, 2019, doi:10.3390/jcm8040538_

Reviewer 1 Report

Review of the manuscript entitled:

“Predictors of response to a medial branch block: MRI analysis of the lumbar spine”

In this study, the authors sought to analyze the associations between radiologic spinal pathology and the response to medial branch blocks (MBB). To this end, they analyzed 252 patients who had undergone MBB during a five-year time span at their institution. 165 patients were eligible for further analysis. The MRI characteristics as well as the response to MBB after one month was assessed. 

While this is an interesting subject, I have some points of criticism:

There has already been some similar research in recent years, which failed to establish a distinct association between the response to MBB and radiologic findings. Even if such an association would be established, this would not necessarily mean not to perform MBB in patients with higher degenerative changes, only because the probability of success might be somewhat smaller.

Further the association between radiologic findings and response found in this study seems to be rather weak. Therefore, the study results do not necessarily lead to the conclusion not to perform MBB in patients with higher degenerative changes.

On the other hand, the authors claim that their results are clinically relevant and that MBB in patients with degenerative disc disease L5/S1 or facet tropism L3/4 is likely to lead to a poor response.

In my view this statement is too strong. It might be adequate to state that MBB in patients with degenerative disc disease L5/S1 or facet tropism L3/4 is somewhat more likely to lead to a poor response.

The authors do not give details on how exactly the MBBs are performed. Moreover, they do not indicate which factors exactly were included in the univariate and in the multivariate logistic regression.

The results of MBBs are only reported as “response” or “no response”. Maybe the statistics could be improved, if a correlation between the percentage of improvement on the NRS scale would be correlated to radiological findings. 

Introduction:

P2 L66

“However, there are only a few studies that factored in the response to a facet joint block or medial branch block (MBB) [22-24].”

Please rephrase. In the discussion the same phrase is used with partly different references. 

Material and Methods:

P3 L91

“…..according to the standard routine performed in our centre.”

The technique should be explained more in detail? Were the blocks performed according to SIS guidelines? 

P3L93f

“The injectate was consisted of 1 ml of 1% lidocaine with 1 mg of triamcinolone and injected at each level. Each 94 injection was performed using intermittent fluoroscopic needle tip guidance.“

What was the total volume of the injectate at one level/ one side? 

P4 L 144 ff

“The most relevant factors associated with a successful response at 1 month post-MBB were included in a univariate logistic regression analysis. The inclusion of variables into the final multivariate logistic regression analysis to evaluate independent factors associated with a successful response at 1 month post-procedure was based on biological plausibility, clinical importance and statistical considerations (P < 0.10).”

Which factors exactly were included in the univariate and in the multivariate logistic regression? 

P8L255

However, there are fewer studies that factored in the response to a facet joint block or MBB [16,19,22,26]

Please rephrase.

P9L278

“In patients with facet tropism at the L3–4 level, the response to MBB for facet joint pain may be rather poor after 1 month.”

I am not sure whether this effect is really so pronounced. 

“When performing MBB, we can expect a poor response in patients with facet tropism at the L3–4 level, and a good response in patients with degenerative disc disease at the L5–S1 level.“

This statement is not entirely supported by the data.

Author Response

Response to reviewers

To Reviewer 1 

In this study, the authors sought to analyze the associations between radiologic spinal pathology and the response to medial branch blocks (MBB). To this end, they analyzed 252 patients who had undergone MBB during a five-year time span at their institution. 165 patients were eligible for further analysis. The MRI characteristics as well as the response to MBB after one month was assessed.

While this is an interesting subject, I have some points of criticism: 

There has already been some similar research in recent years, which failed to establish a distinct association between the response to MBB and radiologic findings. Even if such an association would be established, this would not necessarily mean not to perform MBB in patients with higher degenerative changes, only because the probability of success might be somewhat smaller.

Further the association between radiologic findings and response found in this study seems to be rather weak. Therefore, the study results do not necessarily lead to the conclusion not to perform MBB in patients with higher degenerative changes.

On the other hand, the authors claim that their results are clinically relevant and that MBB in patients with degenerative disc disease L5/S1 or facet tropism L3/4 is likely to lead to a poor response.

In my view this statement is too strong. It might be adequate to state that MBB in patients with degenerative disc disease L5/S1 or facet tropism L3/4 is somewhat more likely to lead to a poor response.

Response: We agreed with your opinion. As you pointed out, we did not mean “not to perform MBB in patients with higher degenerative changes”. We just meant that the response of MBB seem to be poor in patient with L3-4 facet tropism or to be good in patient with L5-S1 disc degeneration. 

Lumbar facet joints syndrome constitutes a common source of pain and remain a misunderstood, misdiagnosed and inadequately treated. In clinical practice, it is accepted that a response to blockade of either the facet joints themselves, or, more frequently, the medial branches is the most reliable means for diagnosing facet joints as the cause of low back pain. However, the response of medial branch block (MBB) is often highly variable. Therefore, we thought that it is necessary to study whether the structure changes of spine such as facet joint or intervertebral disc, not the disease itself, could affect the treatment response. Although the results of this study may not be conclusive, it may be helpful to predict the outcome of MBB in patient with facet tropism or disc degeneration. 

The authors do not give details on how exactly the MBBs are performed. Moreover, they do not indicate which factors exactly were included in the univariate and in the multivariate logistic regression.

Response: Thanks for your valuable suggestion. We apologize about the inadequate description. As you pointed out, we will add the detail on how exactly the MBBs are performed. In this study, all MBB was performed by the standard routine performed in our centre and it was same to other center. We will add the reference in Method section in the revised manuscript (page 3, line 92 – 100). 

In univariate and multivariate analysis, we included the factor only described the table 3.

The inclusion of variables into the univariate analysis were based on biological plausibility, clinical importance and statistical considerations in Table 1. In univariate logistic regression analysis, facet angel difference at L3 – 4 and L5 – S1, disc height at L3 – 4 and L5 – S1, facet fluid at L5 – S1 were analyzed. The inclusion of variables into the final multivariate logistic regression analysis to evaluate independent factors associated with a successful response at 1 month post-procedure was based on biological plausibility, clinical importance and statistical considerations (P < 0.10). In addition, we confirmed that there was no multicollinearity between the variables. This was described in the Method section of previous manuscript. Finally, only facet angel difference at L3 – 4, disc height at L5 – S1, and facet fluid at L5 – S1 were analyzed in multivariate logistic regression analysis. We will add it in the revised manuscript (page 6, line 185 – 191). 

The results of MBBs are only reported as “response” or “no response”. Maybe the statistics could be improved, if a correlation between the percentage of improvement on the NRS scale would be correlated to radiological findings.

Response: Thanks for your valuable question. This issue is an important point. As your comment, it seems to be better to evaluate a correlation between the percentage of improvement on the NRS and structural change of spine. However, the definition of successful responder used in our study is widely used in other studies. Moreover, in clinical practice, a response to blockade the medial branches is known to be reliable means for diagnosing facet joints. Taken these considerations, binary outcome such as responder or non-responder is likely to more compatible in suspicion of facet joint pain than continuous outcome such as NRS change. Therefore, we used binary outcome for our study. 

1. Introduction:

P2 L66

However, there are only a few studies that factored in the response to a facet joint block or medial branch block (MBB) [22-24].”

Please rephrase. In the discussion the same phrase is used with partly different references.

Response: We apologize about the unclear description and your confusion. As you pointed out, we will rephrase it in the discussion section of revised manuscript (page 8, line 269). 

2. Material and Methods:

P3 L91

“…..according to the standard routine performed in our centre.”

The technique should be explained more in detail? Were the blocks performed according to SIS guidelines?

Response: We apologize about the unclear description. According to your valuable opinion, we have added the detail of MBB. In this study, all MBB was performed by the standard routine performed in our centre and it was same to other center. We will add the reference in method section in the revised manuscript (page 3, line 92 – 100). 

3. P3L93f

“The injectate was consisted of 1 ml of 1% lidocaine with 1 mg of triamcinolone and injected at each level. Each injection was performed using intermittent fluoroscopic needle tip guidance.“

What was the total volume of the injectate at one level/ one side?

Response: Thanks for your valuable question. We apologize about the unclear description and your confusion. In case of suspicion to facet joint syndrome, we routinely performed bilateral MBB of the L2, L3, L4 and L5 dorsal rami. The injectate was consisted of 1 ml of 1% lidocaine with 1 mg of triamcinolone and injected at each level. Therefore, the total volume of the injectate was 8 ml in each patient. We add it in the revised manuscript (page 3, line 99 – 100). 

4. P4 L 144 ff

The most relevant factors associated with a successful response at 1 month post-MBB were included in a univariate logistic regression analysis. The inclusion of variables into the final multivariate logistic regression analysis to evaluate independent factors associated with a successful response at 1 month post-procedure was based on biological plausibility, clinical importance and statistical considerations (P < 0.10).” 

Which factors exactly were included in the univariate and in the multivariate logistic regression?

Response: We apologize about the unclear description and your confusion. According to your comments, we add clear sentence to explain the inclusion variables into univariate and multivariate analysis. 

In univariate and multivariate analysis, we included the factor only described the table 3.

The inclusion of variables into the univariate analysis were based on biological plausibility, clinical importance and statistical considerations in Table 1. In univariate logistic regression analysis, facet angel difference at L3 – 4 and L5 – S1, disc height at L3 – 4 and L5 – S1, facet fluid at L5 – S1 were analyzed. The inclusion of variables into the final multivariate logistic regression analysis to evaluate independent factors associated with a successful response at 1 month post-procedure was based on biological plausibility, clinical importance and statistical considerations (P < 0.10). In addition, we confirmed that there was no multicollinearity between the variables. This was described in the Method section of previous manuscript. Finally, only facet angel difference at L3 – 4, disc height at L5 – S1, and facet fluid at L5 – S1 were analyzed in multivariate logistic regression analysis. We will add it in the revised manuscript (page 6, line 185 – 191). 

5. P8L255

However, there are fewer studies that factored in the response to a facet joint block or MBB [16,19,22,26] 

Please rephrase.

Response: We apologize about the unclear description and your confusion. As you pointed out, we will rephrase it in the discussion section (page 8, line 269). 

6. P9L278

In patents with facet tropism at the L3–4 level, the response to MBB for facet joint pain may be rather poor after 1 month.”

I am not sure whether this effect is really so pronounced.

Response: Thanks for your valuable suggestion. This issue is very important, and we agree with you that this should be clarified. As your comment, we change the sentence that facet tropism at the L3 – 4 level may be associated with poor response of MBB after 1 month (page 9, line 292). 

7. When performing MBB, we can expect a poor response in patients with facet tropism at the L3–4 level, and a good response in patients with degenerative disc disease at the L5–S1 level.“

This statement is not entirely supported by the data.

Response: Thanks for your valuable suggestion. This issue is very important, and we agree with you that this should be clarified. As your comment, we change the sentence that degenerative disc disease at the L5–S1 level may be associated with good response of MBB after 1 month (page 9, line 293 – 295).

Reviewer 2 Report

Overall interesting study, presented well and generally well written. I have a few comments

Generally,

- Was there any reason why the blocks were limited to L3-S1 and did not include higher lumbar levels?

- Since you discussed a few number approaching significance and considering your numbers you should add to the limitation the potential for lack of study power.

- The authors could consider assessing the effects of a combination of degenerative changes at the same level in patient and their corresponding response (ie angle changes, the presence and the and reduced height at the same level and the response to a block)

Specifically,

Abstract - the facet angle was not statistically significant p-value >0.05 but the way it is currently worded in the abstract makes it sound like it is.

Introduction and discussion - You mention "joint complex, including the fibrous capsule, synovial membrane, hyaline cartilage and bone [6]. Lumbar facet arthropathy is a common radiographic finding and is suggested to account for between 5% and 90% of low back pain cases [1,8,9]." 

5-90% is a vey wide range virtually representing 0-100%? What does this mean? How clinically relevant is this?

Methods - What is the NRS scale, and where did it come from? A citation would be useful? the citation provided talks about a pain range from 0-10, why did you grade based on 11 points? Also how is this the same or different from a visual analog pain scale or other pain scales?

- why was the facet angle calculated? To compare one side with the other? What if both sides were equally degenerated -- wouldn't this result in a normal angle difference?

- has the angle measurement of the facet joints you used been previously reported? Were the reading measured by one person? who measured it (eg. surgeon, anesthesiologist, radiologist, medical student, fellow etc...)

- has an inter- intra- rater reliability been previously reported? If not this should be listed as a limitation to your methods.

- your method of spondylolisthesis characterization by percentages should be cited, I believe this is the Meyerding classification.

Discussion - this sentence required clarification "In contrast to disc degeneration at L5–S1, we found that the MBB response in patients with facet tropism at L3–4 was poor."

Author Response

Response to reviewers

To reviewer #2

Generally, 

- Was there any reason why the blocks were limited to L3-S1 and did not include higher lumbar levels?

Response: Thanks for your valuable question. In this study, we analyzed the patients who performed MBB due to lower back pain and who suspected of facet joint syndrome because of axial pain, absence of radiculopathy and paravertebral tenderness. Because we enrolled the patient with lower back pain, we performed bilateral MBB at only L2, L3, L4 and L5 dorsal rami, not at upper level of lumbar spine.   

- Since you discussed a few number approaching significance and considering your numbers you should add to the limitation the potential for lack of study power.

Response: This issue is very important, and we agree with your opinion. We have added limitation in the discussion section (page 9, line 286 – 288) 

- The authors could consider assessing the effects of a combination of degenerative changes at the same level in patient and their corresponding response (ie angle changes, the presence and the and reduced height at the same level and the response to a block)

Response: Thanks for your valuable question. In our study, multivariate logistic regression analysis was performed to evaluate the response of MBB and structural change of vertebrae. All situation as a combination of degenerative changes at the same level was also analyzed. Accordingly, facet angle difference at L3–4 and disc height at L5–S1 may be independent factors associated with a successful response to MBB in our study. 

Because our result is that facet tropism and dis degeneration at different level may associated with response to MBB. Further study is likely needed for a combination of degenerative changes at the same level. 

Specifically, 

1. Abstract - the facet angle was not statistically significant p-value >0.05 but the way it is currently worded in the abstract makes it sound like it is.

Response: Thanks for your valuable question. We apologize about the unclear description and your confusion. We will clarify it (page 1, line 29 – 31). 

In univariate logistic regression analysis, the disc height at L5 – S1 and facet angle difference at L3 – 4 were lower in the positive responders (P=0.022 and P=0.087, respectively). In multivariate logistic regression analysis, facet angle difference at L3 – 4 and disc height at L5 – S1 were independent factors associated with a successful response (odds ratio = 0.948; P = 0.038 and odds ratio = 0.864; P = 0.038, respectively). 

2. Introduction and discussion - You mention "joint complex, including the fibrous capsule, synovial membrane, hyaline cartilage and bone [6]a. Lumbar facet arthropathy is a common radiographic finding and is suggested to account for between 5% and 90% of low back pain cases [1,8,9]."

5-90% is a very wide range virtually representing 0-100%? What does this mean? How clinically relevant is this?

Response: We agreed with your opinion. We apologize about the unclear description and your confusion. We will change it (page 2, line 48 – 49). 

3. Methods - What is the NRS scale, and where did it come from? A citation would be useful? the citation provided talks about a pain range from 0-10, why did you grade based on 11 points? Also how is this the same or different from a visual analog pain scale or other pain scales?

Response: Thanks for your valuable question. And we will add a citation about NRS scale (page 3, line 105). 

An 11-point numeric rating scale (NRS) is widely used scale measure pain intensity assessment. NRS is 11-points scale because it's 0 to 10 instead of 1 to 10. 

The three most commonly used scales to assess pain intensity are the Visual Analogue Scale (VAS), the Numerical Rating Scale (NRS), and the Verbal Rating Scale (VRS). Jensen MP, Karoly P. Self-report scales and procedures for assessing pain in adults. In: Turk DC, Melzack R, eds. Handbook of Pain Assessment. 2nd ed. New York: Guilford Publications; 2001:15-34. 

Because of failure rate between these measures, NRS or VRS is favored over VAS. Moreover, the potential benefits of standardizing pain intensity assessment for comparing between studies, clinicians and researchers consider first using the 0-10 NRS over other pain intensity measures. Dworkin RH, Turk DC, Farrar JT, et al. Core outcome measures for chronic pain clinical trials:  IMMPACT recommendations. Pain 2005; 113:9-19. 

4. -why was the facet angle calculated? To compare one side with the other? What if both sides were equally degenerated -- wouldn't this result in a normal angle difference?

Response: We apologize about the unclear description and your confusion. You are right. Facet angle was measured to evaluate facet angle difference. If facet angle show difference between right and left side, it is called facet tropism. 

We analyzed both the facet angle and facet angle difference. As you pointed out, both facet angle was equally degenerated or changed and facet angle difference could be zero. For that reason, we analyzed not only facet angle difference but also both side of facet angle. 

5.- has the angle measurement of the facet joints you used been previously reported? Were the reading measured by one person? who measured it (eg. surgeon, anesthesiologist, radiologist, medical student, fellow etc...)

Response: We appreciate your good suggestion. It is really valuable. 

We measured the facet angle by method previous reported. Chadha M, Sharma G, Arora SS, Kochar V: Association of facet tropism with lumbar disc herniation. Eur Spine J 22:1045–1052, 2013

As your concern, we add the reference of measurement (page 3, line 123). 

In our study, facet angle measured by two experienced pain physicians.

We add the reference in the revised manuscript (page 4, lune 139 – 142). 

6.- has an inter- intra- rater reliability been previously reported? If not this should be listed as a limitation to your methods.

Response: This issue is an important point. As your comment, this was our limitation. All imaging parameters such as facet angle, facet angle difference, facet degeneration grade, facet fluid, spondylolisthesis, and dis height of each patient were measured three times, and the means of the results was calculated. We add it in the revised manuscript. Inter-rater reliability was not reported. We agreed with your opinion and we have added it in the limitation (page 9, line277 – 279).   

7. - your method of spondylolisthesis characterization by percentages should be cited, I believe this is the Meyerding classification.

Response: You are right. As you pointed out, we have added a reference of spondylolisthesis (page 4, line 130). 

8. Discussion - this sentence required clarification "In contrast to disc degeneration at L5–S1, we found that the MBB response in patients with facet tropism at L3–4 was poor."

Response: We apologize about the unclear description and your confusion. As you pointed out, we have changed the sentence (page 8, line 242 – 243).

Round  2

Reviewer 1 Report

Review of the revised version of the manuscript entitled:

“Predictors of response to a medial branch block: MRI analysis of the lumbar spine”

The manuscript has improved in the new version. However, the wording in some instances is still too strong. Moreover I would have appreciated an analysis of the correlation between percentage of improvement on the NRS scale and radiological findings. This does not preclude a binary outcome analysis and may give additional information.

P8 L242/243

“We found that the MBB response in patients with facet tropism at L3–4 was poor, unlike the MBB response in patients with disc degeneration at L5–S1 was good.”

Maybe use: “ was poorer than in patients without facet tropism at L3–4  “ or “facet tropism at L3–4  was acssociated with a poorer response to…”

Likewise.

“Disc degeneration at L5–S1 was associated with a better response..”

P9 L 289/290

“When performing MBB, we are likely to expect a poor response in patients with facet tropism at the L3–4 level, and a good response in patients with degenerative disc disease at the L5–S1 level.”

Better use:

“poorer response in patients with facet tropism at the L3–4 level, and a better response in patients with degenerative disc disease at the L5–S1 level”

Author Response

Response to reviewers

To Reviewer 1 

Predictors of response to a medial branch block: MRI analysis of the lumbar spine” 

The manuscript has improved in the new version. However, the wording in some instances is still too strong. Moreover I would have appreciated an analysis of the correlation between percentage of improvement on the NRS scale and radiological findings. This does not preclude a binary outcome analysis and may give additional information.

Response: We really appreciate your comments. We agreed with your opinion. As you recommended, we changed the sentence. Furthermore, we analyze the correlation between NRS difference or change and radiological findings (Supplement Table 1). Spearman's rank correlation analysis reveal a weak but statistically significant correlation between NRS difference, NRS change and disc height at L5 – S1 (r = -0.209; P = 0.017 and r = -0.206; P = 0.019, respectively; Supplement Table 1) Aslo, there was a marginal statistically significant correlation between NRS change and facet angle difference at L3 – 4 (r = 0.163; P = 0.062). This result was similar to the result of multivariate logistic regression analysis performed using responder group. In clinical practice, a response to MBB is reliable means for diagnosing facet joints pain. Taken these considerations, binary outcome such as responder or non-responder may be more compatible to analyze the associations between radiologic spinal pathology and the response to MBB rather than continuous outcome such as NRS difference or NRS change. We described it in the Method, Result, Discussion section in the revised manuscript. (page 5; line 156 – 157, page 6; line 197 – 201, page 8; line 267 – 273). 

P8 L242/243 

We found that the MBB response in patients with facet tropism at L3–4 was poor, unlike the MBB response in patients with disc degeneration at L5–S1 was good.” 

Maybe use: “ was poorer than in patients without facet tropism at L3–4  “ or “facet tropism at L3–4  was acssociated with a poorer response to…” 

Likewise. 

Disc degeneration at L5–S1 was associated with a better response..”

Response: We agreed with your opinion and we have changed the sentence. (page 8; line 246 – 247) 

P9 L 289/290 

When performing MBB, we are likely to expect a poor response in patients with facet tropism at the L3–4 level, and a good response in patients with degenerative disc disease at the L5–S1 level.” 

Better use: 

poorer response in patients with facet tropism at the L3–4 level, and a better response in patients with degenerative disc disease at the L5–S1 level”

Response: Thanks for your valuable suggestion. And we have changed the sentence. (page 9; line 301 – 302)
